# *Prevotella melaninogenica*, a Sentinel Species of Antibiotic Resistance in Cystic Fibrosis Respiratory Niche?

**DOI:** 10.3390/microorganisms9061275

**Published:** 2021-06-11

**Authors:** Claudie Lamoureux, Charles-Antoine Guilloux, Elise Courteboeuf, Stéphanie Gouriou, Clémence Beauruelle, Geneviève Héry-Arnaud

**Affiliations:** 1Univ Brest, INSERM, EFS, UMR 1078, GGB, 29200 Brest, France; claudie.lamoureux@univ-brest.fr (C.L.); charlesantoine.guilloux@univ-brest.fr (C.-A.G.); e.courteboeuf@orange.fr (E.C.); stephanie.gouriou@univ-brest.fr (S.G.); clemence.beauruelle@univ-brest.fr (C.B.); 2Department of Bacteriology, Virology, Hospital Hygiene, and Parasitology-Mycology, Brest University Hospital, 29200 Brest, France; 3Brest Center for Microbiota Analysis (CBAM), Brest University Hospital, 29200 Brest, France

**Keywords:** antibiotics susceptibility, *Prevotella melaninogenica*, cystic fibrosis, resistance

## Abstract

The importance and abundance of strict anaerobic bacteria in the respiratory microbiota of people with cystic fibrosis (PWCF) is now established through studies based on high-throughput sequencing or extended-culture methods. In CF respiratory niche, one of the most prevalent anaerobic genera is *Prevotella*, and particularly the species *Prevotella melaninogenica.* The objective of this study was to evaluate the antibiotic susceptibility of this anaerobic species. Fifty isolates of *P. melaninogenica* cultured from sputum of 50 PWCF have been included. Antibiotic susceptibility testing was performed using the agar diffusion method. All isolates were susceptible to the following antibiotics: amoxicillin/clavulanic acid, piperacillin/tazobactam, imipenem and metronidazole. A total of 96% of the isolates (48/50) were resistant to amoxicillin (indicating beta-lactamase production), 34% to clindamycin (17/50) and 24% to moxifloxacin (12/50). Moreover, 10% (5/50) were multidrug-resistant. A significant and positive correlation was found between clindamycin resistance and chronic azithromycin administration. This preliminary study on a predominant species of the lung “anaerobiome” shows high percentages of resistance, potentially exacerbated by the initiation of long-term antibiotic therapy in PWCF. The anaerobic resistome characterization, focusing on species rather than genera, is needed in the future to better prevent the emergence of resistance within lung microbiota.

## 1. Introduction

The importance and abundance of strict anaerobic bacteria in the respiratory microbiota of people with cystic fibrosis (PWCF) is now established. These recent data are based on high-throughput sequencing [1,2] and cultural techniques [3,4,5]. Anaerobes colony count has been evaluated between 1.10^4^ to 9.10^7^ colony forming units per milliliter in sputum culture [3]. Numerous anaerobic genera have been identified as part of the CF core lung microbiota such as *Prevotella, Veillonella, Porphyromonas, Fusobacterium,* or *Peptostreptococcus* [2,6,7]. However, these bacteria remain the unknowns of the lung, both in terms of diversity, resistance or impact on the pathophysiology of CF disease [8]. One of the most prevalent and abundant anaerobic genera described in CF lung is *Prevotella*, and particularly the species *Prevotella melaninogenica* [4,5,9]. Contradictory hypotheses on the potential role of *Prevotella* genus have been previously developed [8]. On the one hand, it has been associated with better lung function and less inflammation [2,10,11,12]. On the other hand, *Prevotella* species are able to enhance other bacteria’s pathogenicity in lung [13], to produce pro-inflammatory short chain fatty acids [14] and are described to be resistant to antibiotics [15]. Focusing on this last point, previous studies have reported resistance acquisition of *Prevotella* isolates, due to the production of enzyme (mainly beta-lactamase [13,16,17]) or the carrying of the resistance gene [18,19]. Moreover, in CF, an over-expression of antibiotic resistance may be observed due to several reasons: repeated or chronic use of antimicrobial treatment, administration of high doses of antibiotic therapy, or prescription of antibiotic molecules at sub-clinical doses for anti-inflammatory properties. As the issue of antibiotic resistance is crucial, even more in PWCF, antibiotic susceptibility monitoring of this main anaerobic genus is necessary.

The objective of this study was to evaluate the antibiotic resistance rate of *Prevotella melaninogenica*, the most prevalent anaerobic species of the bronchopulmonary microbiota of PWCF.

## 2. Materials and Methods

### 2.1. Patients Characteristics

Fifty PWCF were included in the study. The following sociodemographic and clinical parameters were recorded: age, sex, *CFTR* (CF transmembrane conductance regulator) gene mutation, lung function, diabetes, chronic antibiotics administration (azithromycin, aztreonam, colistin, and tobramycin), and antibiotic administration one month before the sample. The patients’ lung involvement was assessed by the measurement of the maximum expiratory volume per second (FEV_1_) by spirometry tests, which was defined by three stages: early (FEV_1_ > 70%), intermediate (40 < FEV_1_ < 70%) and advanced (FEV_1_ < 40%) [20].

### 2.2. Clinical Isolates and Antibiotic Susceptibility Testing

Fifty isolates of *Prevotella melaninogenica* cultured from sputum of 50 PWCF collected at the Western Brittany CF center (Roscoff, France) have been included in this study. Cultural methods from sputum (three different media, 21 days of culture at 37 °C in anaerobic atmosphere) have been previously described [5]. These 50 isolates have been identified by matrix-assisted laser desorption/ionisation time-of-flight mass spectrometry (MALDI-TOF MS Biotyper MBT) (Bruker, Billerica, MA, USA); a score of ≥ 2.0 was considered as accurate species-level identification; a score ≥ 1.7 but < 2.0 as a probable genus-level identification; a score < 1.7 as “unidentified”. In order to improve isolates identification, 1 μL of 70% formic acid LC/MS (VWR, Radnor, PA, USA) was added before the addition of 1 μL of portioned IVD-HHCA matrix (Bruker, Billerica, MA, USA) [5]. Antibiotic susceptibility testing was performed using the agar diffusion method according to the recommendations of EUCAST/CASFM 2019 [21]. From a fresh culture, an inoculum of 1 McFarland was prepared in physiological water (ThermoFisher, Waltham, MA, USA). The bacterial suspensions were inoculated onto Brucella agar plates supplemented with vitamin K1, hemin (Sigma-Aldrich, Dorset, England) and sheep blood (ThermoFisher, Waltham, MA, USA). The following antibiotics were tested (ThermoFisher, Waltham, MA, USA; Bio-Rad, Hercules, CA, USA): amoxicillin (20 μg)/clavulanic acid (10 μg), clindamycin (2 μg), imipenem (10 μg), metronidazole (4 μg), moxifloxacin (5 μg), and piperacillin (30 μg)/tazobactam (10 μg). Susceptibility to amoxicillin was assessed by determining the minimum inhibitory concentration (MIC) using an E-test strip (BioMérieux, Marcy-l’Etoile, France). Agars were then incubated under anaerobic conditions at 37 °C for 48 h (Bactron^®^ anaerobic chamber, Sheldon manufacturing, Cornelius, NC, USA). *P. melaninogenica* isolates with a MIC for amoxicillin greater than or equal to 0.5 mg/L were classified as beta-lactamase producer [21]. Isolates resistant to at least one agent in three or more antimicrobial categories were categorized as multidrug-resistant (MDR) [22].

### 2.3. Statistical Analysis

Statistical analysis was performed using the Chi-square and Fisher tests. The alpha risk has been reported significant when it was less than 0.05.

## 3. Results

### 3.1. Patients Characteristics

The study population was composed of 52.0% of males (26/50) and the median age was 28 years (range: 6–58). Subject characteristics (demographic, clinical and biological) are outlined in Table 1. The stage of lung involvement was early for 22% (11/50), intermediate for 52% (26/50) and advanced for 26% (13/50) of PWCF.

### 3.2. Antibiotic Susceptibility Testing

Antibiotic susceptibilities of *P. melaninogenica* isolates are presented in Table 2. All isolates were susceptible to the following antibiotics: amoxicillin/clavulanic acid, piperacillin/tazobactam, imipenem, and metronidazole. A total of 96% of *P. melaninogenica* isolates (48/50) were classified as beta-lactamase producers (MIC for amoxicillin > 0.5 mg/L), and 38% of these isolates (18/48) had a MIC for amoxicillin superior to 256 mg/L (Figure 1). The MIC for amoxicillin for the two non-beta-lactamase producers isolates have been evaluated at 0.094 mg/L and inferior to 0.016 mg/L (Figure 1). A total of 34% (17/50) and 24% (12/50) of isolates were resistant to clindamycin and moxifloxacin, respectively. Ten percent of isolates (5/50) were categorized MDR, due to a combined resistance to amoxicillin, clindamycin and moxifloxacin.

### 3.3. Associations between Antibiotic Resistance and Antibiotic Administration

Amoxicillin, clindamycin and moxifloxacin resistance percentages were compared to antibiotic administration one month before the sample (oral or intravenous: yes or no) and chronic antibiotic administration (azithromycin, aztreonam, colistin or tobramycin: yes or no). A significantly positive correlation was found between clindamycin resistance and oral chronic azithromycin administration (*p* = 0.002, Chi-square test).

## 4. Discussion

The potential impact of strict anaerobic bacteria in the pathophysiology of CF disease and in the respiratory function of PWCF is still at the hypothesis stage. The important and now undeniable place of these bacteria within the CF respiratory microbiota led to the study of their antibiotic susceptibilities to better understand the issue of resistance. For the genus *Prevotella*, disparities in resistance rates have been observed among the different studies but all the results agree on an increase in antibiotic resistance from clinical isolates in CF, as well as in other diseases [15,23].

Concerning beta-lactams, resistance is mainly due to the production of beta-lactamases, which induce a loss of susceptibility to penicillin, and that of the first, second, and third (oral) generation cephalosporins [21]. Several beta-lactamase detection methods can be used, combining phenotypic (mostly performed in routine practice) and molecular (*cfxA* resistance genes detection approaches [18,19]). Indeed, the detection of *cfx*A genes in *Prevotella* isolates has been associated with high MICs of amoxicillin [19,24]. However it remains important to determine if the isolates are functionally resistant to amoxicillin [25]. In this study focusing on *P. melaninogenica*, a high percentage of isolates (96%) were categorized as beta-lactamase producers using phenotypic detection based on the MIC of amoxicillin determination. This result is slightly higher than previous results for this species (Table 3) which can be explained by three main reasons. Firstly, the other studies used the nitrocefin test for phenotypic detection of beta-lactamase. However the recommendation of EUCAST/CASFM 2019 [21] is to avoid nitrocefin, considered as a poor substrate which may underestimate beta-lactamase activity detection. Moreover interference with the isolates pigmentation (as *P. melaninogenica*) represents a limit to this test based on a colorimetric reaction [26]. As formerly demonstrated, the determination of the MIC of amoxicillin is a more reliable method of detection [27]. Secondly, this study focused only on *P. melaninogenica* species while others’ studies are based on several *Prevotella* species, for which resistance rates may differ [26]. Thirdly, in this study, isolates have been obtained from sputum of PWCF, for which antibiotic selective pressure is higher due to chronic antibiotic therapy administration and may lead to diminution of antibiotic susceptibility [15]. Moreover, CF lung has been described as a potential reservoir of resistance genes (as oral sphere) because *cfx*A genes coding beta-lactamase could be transferred among bacteria by horizontal transfer gene mechanisms [8,28,29]. Fourthly, as demonstrated for other genera known to carry *cfx*A genes (*Capnocytophaga*, *Bacteroides*,…), the expression of beta-lactamase may differ according to major substitutions in the *cfx*A gene sequence or to the influence of the genetic environment of *cfx*A [30]. This molecular part has not been evaluated in this study.

For the other beta-lactams, all of the isolates of the present study were susceptible to piperacillin/tazobactam, amoxicillin/clavulanic acid, and imipenem (Table 4). No resistance have been described to date for amoxicillin/clavulanic acid and imipenem, however resistance to piperacillin/tazobactam was previously reported in the literature for *Prevotella* spp. [3,31].

As beta-lactams, emerging resistance to macrolides and fluoroquinolone has been described for *Prevotella* spp. [42,43]. In our study, 24% of *P. melaninogenica* isolates were resistant to moxifloxacin, which was concordant with the results of Ulger Toprak et al. [26] (Table 4). Thirty-four percent of *P. melaninogenica* isolates were resistant to clindamycin, which was higher than previously described in non-CF individuals, as no resistance data are available for PWCF [18,26,32]. Chronic azithromycin administration, prescribed at sub-clinical doses for anti-inflammatory properties, was significantly correlated with clindamycin resistance (*p* < 0.05, Chi-square test), as previously highlighted in PWCF [16]. This result is consistent with the fact that one of the main factors leading to antibiotic resistance identified for anaerobic bacteria is long-term antibiotic use [43,44]. Sherrard et al. [15] showed that the administration of beta-lactam or tetracycline in the year prior to pulmonary sample collection in PWCF was associated with higher MICs of amoxicillin, ceftazidime and tetracycline for *Prevotella* spp. In the future, the worsening resistance of the *Prevotella* genus is a threat and should be monitored. Firstly, different studies highlighted the increase of MDR anaerobic isolates. Ulger Toprak et al. [26] have demonstrated that nearly 10% of *Prevotella* isolates were categorized MDR (antibiotics involved: ampicillin, clindamycin, moxifloxacin and tetracycline). In our study, a similar rate of MDR isolates was determined (five isolates of *P. melaninogenica* resistant to amoxicillin, moxifloxacin and clindamycin). Secondly, in addition to beta-lactamase activity, *Prevotella* species are able to produce extended-spectrum beta-lactamase (ESBLs), associated with treatment failure with cephalosporin antibiotics. In a study focused on *Prevotella* isolates isolated from respiratory samples of PWCF, 76% were found to produce ESBLs [17]. This resistance is rarely investigated in routine practice, as in our study, and may be underestimated. Finally, resistance to metronidazole, an anti-anaerobic molecule, has been reported for *P. melaninogenica* [18] and *Prevotella* spp. [31,42,45,46,47] with a resistance rate under 10% (Table 4). Tunney et al. [3] have observed a higher resistance rate evaluated at 46% in *Prevotella* spp. isolates (including four species: *P. corporis, P. disiens, P. melaninogenica and P. salivae).* No metronidazole resistant isolates of *P. melaninogenica* were described in our study.

For all the antimicrobial categories, results obtained in our study compared with previous ones showed the importance of the species level when considering antibiotic resistance and the difficulty to compare antibiotic resistance from isolates of different species belonging to the same genus [26] (Table 4). The targeted study based on species is therefore the most informative. In routine practice, this approach requires equipment (e.g., anaerobic chamber, collecting device for anaerobic atmosphere preservation) for the detection and identification of strict anaerobic bacteria in CF respiratory samples as well as for the realization of antibiogram.

## 5. Conclusions

This preliminary study of 50 clinical isolates of a predominant species of the lung “anaerobiome” showed a high percentage of resistance and the presence of MDR isolates, potentially exacerbated by the initiation of long-term antibiotic therapy in PWCF. In the future, it will be important to continue to characterize the anaerobic resistome by focusing on species rather than genera in order to better prevent the emergence of resistance within pulmonary microbiota.

## Figures and Tables

**Figure 1 microorganisms-09-01275-f001:**
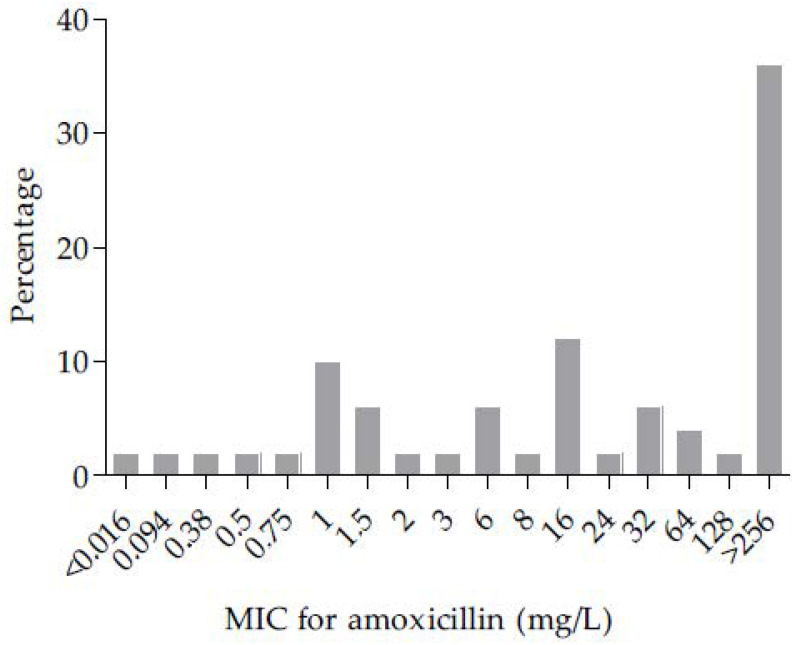
Distribution of minimum inhibitory concentration (MIC) for amoxicillin (mg/L) among the 50 isolates of *Prevotella melaninogenica*.

**Table 1 microorganisms-09-01275-t001:** Study cohort characteristics: demographic, clinical and biological data. A total of 50 people with cystic fibrosis (PWCF) were included.

PWCF Characteristics		Percentage (*n*)
Age group (years)	≤6	4 (2)
	7–13	8 (4)
	14–18	6 (3)
	19–25	28 (14)
	26–30	22 (11)
	≥30	32 (16)
Sex	Female	48 (24)
	Male	52 (26)
p.F508del mutation	Homozygote	60 (30)
	Heterozygote	28 (14)
	Other mutations	12 (6)
Lung function (FEV_1_ %)	≤40	26 (13)
	40–70	52 (26)
	>70	22 (11)
Chronic antibiotic administration	Azithromycin	44 (22)
	Aztreonam	10 (5)
	Colistin	42 (21)
	Tobramycin	20 (10)
Antibiotics administration one month before the sample	Oral *	36 (18)
	Intravenous **	16 (8)

* amoxicillin/clavulanic acid, cefpodoxime, ciprofloxacin, doxycycline, linezolid, minocycline, pristinamycin, trimetoprim/sulfamethoxazole, ** amikacin, aztreonam, ceftazidime, doxycycline, linezolid, meropenem, piperacillin/tazobactam, tobramycin, trimetoprim/sulfamethoxazole, FEV_1_: Forced Expiratory Volume in one second.

**Table 2 microorganisms-09-01275-t002:** Antibiotic susceptibilities of 50 isolates of *Prevotella melaninogenica* (according to recommendations of EUCAST/CASFM 2019 [21].

Antibiotic	Interpretative Categories n (%)	
	S	R
Amoxicillin	2 (4)	48 (96)
Amoxicillin/clavulanic acid	50 (100)	0
Piperacillin/tazobactam	50 (100)	0
Imipenem	50 (100)	0
Clindamycin	33 (66)	17 (34)
Moxifloxacin	38 (76)	12 (24)
Metronidazole	50 (100)	0

S: susceptible, R: resistant.

**Table 3 microorganisms-09-01275-t003:** Studies reporting the percentage of beta-lactamase activity in *Prevotella* spp.

	Beta-Lactamase Activity (%)	Number of Isolates	*Prevotella* Species	Cystic Fibrosis	Method of Detection
This study	96	50	*P. melaninogenica*	Yes	MIC of amoxicillin
Bahar et al., 2005 [32]	68	19	*P. melaninogenica*	No	Nitrocefin test
Fujita et al., 2019 [33]	85	27	5 *Prevotella* species	No	Nitrocefin test
Bancescu et al., 2015 [23]	33	33	5 *Prevotella* species	No	Nitrocefin test
Fernandez et al., 2015 [34]	46	41	2 *Prevotella* species	No	Nitrocefin test
Montagner et al., 2014 [25]	20	15	4 *Prevotella* species	No	Nitrocefin test
Sherrard et al., 2013 [16]	59	157	*Prevotella* spp.	Yes and no	Nitrocefin test
Tran et al., 2013 [35]	75	16	*Prevotella* spp.	No	Nitrocefin test
Kuriyama et al., 2007 [36]	37	499	*Prevotella* spp.	No	Nitrocefin test
Mosca et al., 2007 [37]	18	39	3 *Prevotella* species	No	Nitrocefin test
Iwahara et al., 2006 [38]	31	139	8 *Prevotella* species	No	Nitrocefin test
Behra-Miellet et al., 2003 [39]	65	40	*Prevotella* spp.	No	Nitrocefin test
Dubreuil et al., 2003 [40]	58	100	12 *Prevotella* species	No	Nitrocefin test
Matto et al., 1999 [41]	31	171	3 *Prevotella* species	No	Nitrocefin test

**Table 4 microorganisms-09-01275-t004:** Studies reporting antibiotic resistance of *Prevotella* spp.

Reference	Cystic Fibrosis	*Prevotella* Species	Isolates Identification	Number of Isolates	Location	Percentage of Resistance						
						Amoxicillin or Ampicillin	Amoxicillin/Clavulanic Acid	Piperacillin/Tazobactam	Imipenem	Clindamycine	Moxifloxacine	Metronidazole
This study	Yes	*P. melaninogenica*	Mass spectrometry	50	France	96.0	0	0	0	34.0	24.0	0
Veloo et al., 2019 [18]	No	*P. melaninogenica*	Mass spectrometry	21	Netherlands	66.7	-	-	-	4.8	-	4.8
Ulger Toprak et al., 2018 [26]	No	*P. melaninogenica*	16S rRNA gene sequencing	44	11 countries	67.4	-	-	-	16.3	20.5	-
Bahar et al., 2005 [32]	No	*P. melaninogenica*	Biochemical tests	19	Turkey	-	-	-	0	10.5	-	0
Byun et al., 2019 [45]	No	8 *Prevotella* species	Mass spectrometry	33	Korea	-	-	0	0	45.0	9.0	3.0
Cobo et al., 2019 [31]	No	*Prevotella* spp.	Mass spectrometry	30	Spain	-	0	3.0	0	40.0	30.0	7.0
Bancescu et al., 2015 [23]	No	*5 Prevotella* species	Biochemical tests	33	Romania	33.0	0	-	-	3.0	-	0
Shilnikova et al., 2014 [46]	No	13 *Prevotella* species	Mass spectrometry	42	Russia	-	0	-	0	11.9	-	4.8
Xie et al., 2014 [47]	No	10 *Prevotella* species	16S rRNA gene sequencing	42	China	30.9	-	-	0	38.1	-	9.5
Tran et al., 2013 [35]	No	*Prevotella* spp.	16S rRNA gene sequencing	16	Japon	69.0	-	0	0	19.0	-	0
Seifert et al., 2010 [48]	No	7 *Prevotella* species	Biochemical tests	21	Germany	-	-	-	-	9.5	0	0
Papaparaskevas et al., 2008 [42]	No	10 *Prevotella* species	Biochemical tests	141	Greece	-	-	0	0	31.0	38.0	8.0
Tunney et al., 2008 [3]	Yes	4 *Prevotella* species	Biochemical tests	14	Ireland	36.0	-	7.0	-	36.0	-	46.0
Mosca et al., 2007 [37]	No	3 *Prevotella* species	Biochemical tests	39	Italy	18.0	0	-	-	-	7.7	0
Behra-Miellet et al., 2003 [39]	No	*Prevotella* spp.	Biochemical tests	40	France	65.0	0	-	0	5.0	-	0
Mory et al., 1998 [49]	No	10 *Prevotella* species	Biochemical tests	56	France	25.0	0	-	0	0	-	0

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
