# Peer review of "Prevotella melaninogenica, a Sentinel Species of Antibiotic Resistance in Cystic Fibrosis Respiratory Niche?"

_microorganisms, 2021, doi:10.3390/microorganisms9061275_

Round 1

Reviewer 1 Report

The manuscript of Lamoureux et al. deals with the decetection of antimicrobial resistances of P. melaninogenica strains from the sputa of CF patients. The study design, implementation and standards of communications are all of high level and this way the scientific community can get important information about the AMR of an important anaerobic pathogen in CF patients. One major concern is that the authors found a high beta-lactamase positivity/amoxicillin resistance value that might be supported by cfxA detection too. My suggestion is to further analyse/discuss it as CfxA is the only described beta-lactamase for Prevotellae.

A minor concern is that in the Title, the authors made generalizations for P. melaninogenica but did not compare their results at least in the DIscussion with those of other species.

Author Response

The manuscript of Lamoureux et al. deals with the detection of antimicrobial resistances of P. melaninogenica strains from the sputa of CF patients. The study design, implementation and standards of communications are all of high level and this way the scientific community can get important information about the AMR of an important anaerobic pathogen in CF patients.

1. One major concern is that the authors found a high beta-lactamase positivity/amoxicillin resistance value that might be supported by cfxA detection too. My suggestion is to further analyse/discuss it as CfxA is the only described beta-lactamase for Prevotellae.

We thank the Reviewer 1 (R1) for this relevant suggestion and we agree on the fact that discussion about cfxA genes were missing in the manuscript. In the discussion section concerning beta-lactamase detection in our study, we discussed (adding more references) about cfxA genes transmission and expression focusing on CF lung microbiota (Lines 146-148; Lines 163-169).

2. A minor concern is that in the Title, the authors made generalizations for P. melaninogenica but did not compare their results at least in the discussion with those of other species.

We agree with R1 that we can’t exclude the fact that others Prevotella species could also be sentinel of the antibiotic resistance in CF lung microbiota. In this study, we decide to focus on P. melaninogenica, described as the most prevalent Prevotella species in the CF “anaerobiome”. That’s the reason why we open the resistance hypothesis to others species with a non exclusive title by mentioning “a sentinel species” instead of “the sentinel species”.

Reviewer 2 Report

This manuscript describes antimicrobial resistance in isolates of Prevotella melaninogenica derived from people with CF. The authors demonstrate a high level of resistance and also beta-lactamase production, likely to be associated with long-term antibiotic therapy in these patients. The manuscript is clearly written and would be a useful addition to the current literature. The authors focus on the species-specific characteristics of P. melaninogenica however they do not describe their method of species-level identification which would be an important addition.

Minor comments:

Line 66: It would be useful if the authors could add a sentence or two describing their methods for isolation of these organisms from sputum.

Materials and methods:  It would be important for the authors to add a short section describing the method used to identify these isolates to species level. Is it difficult to separate these species by the chosen method? In lines 151-153 and 191 and in the discussion and conclusion it is stated other studies did not differentiate between the different Prevotella spp. and the authors state that there are likely to be differences between the species with regard to antimicrobial resistance. It is therefore is important to outline the method used in this study.

Line 94: Perhaps it would be better to replace “People with Cystic Fibrosis Characteristics” with “CF patient demographics”.

Line 97: I would suggest replacing “resume” with another word, such as “outlined”.

Table 3 provides a useful comparison of similar studies. It might be helpful to add an additional column with the methods used to identify these organisms in these other studies.

Author Response

This manuscript describes antimicrobial resistance in isolates of Prevotella melaninogenica derived from people with CF. The authors demonstrate a high level of resistance and also beta-lactamase production, likely to be associated with long-term antibiotic therapy in these patients. The manuscript is clearly written and would be a useful addition to the current literature. The authors focus on the species-specific characteristics of P. melaninogenica however they do not describe their method of species-level identification which would be an important addition.

1. Line 66: It would be useful if the authors could add a sentence or two describing their methods for isolation of these organisms from sputum.

This study focusing on the antibiotic resistance of P. melaninogenica follows a descriptive study of the CF “anaerobiome” recently published (Lamoureux et al., https://doi.org/10.1038/s41598-021-85592-w), in which the methods of isolation of anaerobic bacteria are described. As the Reviewer 2 (R2) suggested, we added more details in brief about cultural methods, referring to this publication (Lines 68-69).

2. Materials and methods:  It would be important for the authors to add a short section describing the method used to identify these isolates to species level. Is it difficult to separate these species by the chosen method? In lines 151-153 and 191 and in the discussion and conclusion it is stated other studies did not differentiate between the different Prevotella spp. and the authors state that there are likely to be differences between the species with regard to antimicrobial resistance. It is therefore is important to outline the method used in this study.

We fully agree with R2 about the importance to outline the identification method of anaerobic bacteria. We followed R2’s advice and added more details in the method section to explain briefly the method used for the isolates identification (Lines 69-75). These 50 isolates have been identified using matrix-assisted laser desorption/ionisation time-of-flight mass spectrometry (MALDI-TOF MS Biotyper MBT) (Bruker, Billerica, USA). The identification criteria of MALDI-TOF MS were as follows: a score of ≥2.0 was considered as accurate species-level identification; ≥ 1.7 but < 2.0 as a probable genus-level identification; an isolate with a score < 1.7 was considered as “unidentified”. In order to improve isolates identification, 1 μL of 70% formic acid LC/MS (VWR, Radnor, USA) was added before the addition of 1 μL of portioned IVD-HHCA matrix (Bruker, Billerica, USA) (Lamoureux et al., https://doi.org/10.1038/s41598-021-85592-w).

3. Line 94: Perhaps it would be better to replace “People with Cystic Fibrosis Characteristics”with “CF patient demographics”.

We thank R2 for this remark and we rephrase the section title with “Patient demographics”.

4. Line 97: I would suggest replacing “resume” with another word, such as “outlined”.

As R2 suggested, we replace “resume” by “outlined”.

5. Table 3 provides a useful comparison of similar studies. It might be helpful to add an additional column with the methods used to identify these organisms in these other studies.

We thank R2 for this very relevant suggestion. We added an additional column to detail the identification methods of Prevotella isolates in the others studies.